# An Agricultural Interval Two-Stage Fuzzy Differential Water Price Model (ITS-DWPM) for Initial Water Rights Allocation in Hulin, China

**Shuo Yan [1], Liquan Wang [1,\*] and Tienan Li [2]**

[1] College of Water Resources and Electric Power, Heilongjiang University, Harbin 150080, China; yanshuo31869@163.com

[2] Departments of Heilongjiang Water Conservancy Research Institute, Harbin 150080, China; litienan0019@163.com

\* Correspondence: 15304640067@163.com; Tel.: +86-153-0464-0067

**Abstract:** In recent years, China's agricultural water consumption has been high, while water inefficiency has restricted the development of the economy. In this study, we developed an agricultural interval two-stage fuzzy differential water prices model (ITS-DWPM) by incorporating the techniques of two-stage programming (TP) and interval-parameter programming (IPP) based on the differential water price. The ITS-DWPM can link the associated economic penalty attributed to the violation of the preregulated water target and the total system benefit under limited data availabilities (expressed stochastic and interval values); meanwhile, inaccurate water quantity data and dynamic economic data would be resolved. The methodology tended to resolve the complexity of initial water rights allocation problems, incorporating the relevant sectors as well as the agricultural irrigation system construction involved in water management decision. Furthermore, the model takes into account the restrictions on water quantity and price. In addition, multiple decision results were calculated under different states of water shortage. The developed method is applied to initial water rights allocation and agricultural irrigation system construction. The results generated can assist decision makers not only in formulating water rights allocation strategies, but also in balancing the contradiction between economic objectives and water resources control indicators. In 2020, the irrigation water utilization coefficient of each agricultural irrigation area in Hulin City should reach 0.55. Hutou needs to increase its agricultural irrigation channels by at least 4.49%, Shitouhe by at least 4.03%, and Daxinancha by at least 4.49%.

**Keywords:** initial water rights allocation; differential agricultural water price; water-saving adjustment; interval two-stage programming

## 1. Introduction

The Chinese water crisis is getting more and more serious. China is one of the 13 water-scarce countries in the world, with per capita water consumption accounting for only 25% of the world's per capita water consumption [1]. In 2017, the Chinese effective utilization coefficient for farmland irrigation water was only 0.542, which is far below the global level of 0.7–0.8. The total agricultural water consumption in Heilongjiang Province was 31.644 billion cubic meters, accounting for 89.6% of the province's total water consumption [2]. The paddy field in Heilongjiang Province is vast, and the irrigation water consumption of farmland is huge. Due to the aging and imperfection of farmland water conservancy projects, a large proportion of water resources is wasted. Groundwater resources in some areas are overexploited. To tackle the problem of inefficient agricultural water consumption, Central Document No. 1 has reformed many agricultural areas. One of the areas of comprehensive

agricultural reform is water price, with the goal being to adjust the price of agricultural water and promote water conservation. Again, the rational allocation of water rights is a fundamental aspect of the comprehensive reform of agricultural water prices, and will also play a role in effective water saving [3].

The comprehensive reform of agricultural water price measures is intended to raise the price of water, which is an effective method to guarantee national food and water security. Therefore, it is very important to formulate a reasonable agricultural water price system [4–7]. Dono proposed two methods for the application of agricultural water pricing. The first method, based on the metered use of water by farms, is known as the volumetric pricing method; the second is an area-based pricing method, whereby fees are charged per hectare according to the estimated average water use for each crop [8]. Poor measurement facilities in agricultural irrigation districts lead to poor applicability of traditional water rights allocation methods. Färe used the directional output distance function to derive estimates of production inefficiency, shadow prices for polluting outputs, and the associated pollution costs [9]. Shadow prices reflect the scarcity of resources and the demands for final products in the social economy, but cannot reflect economic changes and changes in demand. Shadow prices could increase the water elasticity demands and production output framework [10,11]. In summary, multiple factors need to be considered in establishing a reasonable agricultural water price system. Differentiated water pricing methods can reflect changes in the social economy and make agricultural water prices more reasonable [12–16].

A number of uncertain optimization methods can be introduced to help water managers face the challenges of uncertainties and their interactions [17–23]. When irrigation meets the water demand, it will produce benefits; and when the demand is not satisfied, losses will occur. Therefore, one type of stochastic programming (SP), named two-stage stochastic programming (TSP), can be used to handle uncertainties expressed as probabilistic distributions. TSP provides an effective link between policies and economic penalties, and has advantages in terms of reflecting the complexities of system uncertainties as well as analyzing policy scenarios when preregulated targets are violated [24]. On the one hand, there are no accurate data; on the other hand, the dynamic changes of influencing factors cannot be predicted. This makes it impossible to deal with ambiguity in the water resources system. Zeng developed a two-stage credibility-constrained programming with a Hurwicz criterion (TCP-CH) approach for water resources management and planning under uncertainty. It can also check for system failure risk based on different risk preferences of decision makers [25]. Incorporation of interval-parameter programming (IPP) within a two-stage stochastic programming (TSP) framework can reflect not only the uncertainties expressed as probability distributions but also interval numbers. Moreover, it can provide an effective link between conflicting economic benefits and the associated penalties handed out for the violation of the predefined policies [26–30]. Previously, a number of researchers have incorporated quadratic programming (QP), inexact credibility-constrained programming (ICP), and fuzzy programming (FP) into a framework for water resources management. For example, Zeng et al. (2015) developed a mixed inexact-quadratic fuzzy water resources management model (IQT-WMMF) for floodplains, incorporating techniques of credibility-constrained programming (CP), two-stage programming (TP), interval-parameter programming (IPP), and QP within a general framework of limited data availability [31]. Unfortunately, few studies have previously focused on the IPP, FP, and TSP methods of sustainable WMF planning within a two-stage context.

Therefore, the objective of this study is to develop an agricultural interval two-stage fuzzy differential water price model (ITS-DWPM) for initial water rights allocation. The developed ITS-DWPM method is an integrated optimization technique for tackling multiple uncertainties expressed as discrete intervals, nonlinearity, and fuzzy sets. The method is applied to a real case study of initial water rights allocation in Hulin City, China. Keeping in mind the uncertain irrigation needs when considering water prices and water demand, the method can improve the practicality and pertinence of the two stages. At the same time, we carried out a policy analysis of the irrigation water utilization coefficient in the initial water rights allocation. At the same time, the impact of the changes on the initial water

rights allocation is quantified. Finally, we couple the results of the two analyses to support irrigation district managers in initial water rights allocation planning and policy adjustment. The method can provide scientific guidance for the initial water rights allocation of each agricultural irrigation district in the region.

## 2. Methodology

Figure 1 presents the framework of ITS-DWPM application in Hulin City. This method can lead to the achievement of the water consumption target when there is uncertain water demand for irrigation, and can also improve the practicability and pertinence of the two stages. Meanwhile, the effect of the utilization coefficient of irrigation water on initial water rights allocation was considered and a risk analysis was carried out by combining uncertainty and the water weight allocation system. Finally, coupled with the results of the two analyses, the results can support irrigation district managers in carrying out initial water rights allocation planning and policy adjustment.

Due to the loss of irrigation and water conservancy projects, a lack of irrigation district management system, a lack of matching terminal metering equipment in irrigation areas, and the low price of agricultural irrigation water, the efficiency of agricultural water use in Hulin is low. Therefore, the wasting of agricultural water can be reduced by optimizing the initial allocation of water rights in this area. Some factors considered include climate, the dynamic development of agriculture, and farmers' income and expenditure; however, the redistribution capacity is limited by the scarcity of water resources, and the fluctuation of water prices and other factors cannot be reflected. This paper optimizes the allocation of water resources through the adjustment of economic leverage, and formulates matching empirical decisions. Due to temporal and spatial differences in water provision and demand, the initial water rights allocation designs will vary under different policy requirements and allocation targets. If the initial water rights allocation aims preregulated by decision makers is too high, the actual water consumption will not be met; a shortage would thus be generated. Correspondingly, the demand must be curtailed with reduced production activities, resulting in a decreased net system benefit (i.e., a penalty due to shortfalls). In addition, if the aims as preregulated by the decision makers are too low, a surplus may be generated (i.e., a wastly due to surpls). We must constantly adjust the allocation of water rights to formulate reasonable agricultural water prices, ultimately finding the optimal allocation of initial water rights for agricultural irrigation in the area, maximizing the system revenue, reducing agricultural water consumption, and promoting agricultural water users 'awareness of water saving measures. The goal of comprehensive reform of agricultural water prices should be taken into account.

In a practical problem of initial water rights allocation, uncertainties may be expressed as random variables, which result in the relevant decisions being made under varying probability levels. Such a problem can be formulated as a two-stage stochastic programming (TSP) model [32]. However, in an initial water rights allocation system, the impacts on differential water price of uncertain benefits and penalty data may cause relevant objective function nonlinearity. Interval programming can deal with the nonlinearities in the objective function and reflect uncertainty, expressed as interval values. Therefore, by introducing ITS into the TSP model, an inexact two-stage programming (ITSP) model can be expressed as follows:

Subject to

$$\text{Max } f^{\pm} = \sum_{i=1}^{m} \left( G_i^{\pm} - P_{0i}^{\pm} \right) \times wc_i^{\pm} - \sum_{i=1}^{m} \left( G_i^{\pm} - P_{1i}^{\pm} \right) \times Yc_i^{\pm} \tag{1}$$

$$P_{1i}^{\pm} \leq P_{0i}^{\pm} \leq P_{\max}$$

$$P_{1i}^{\pm} \leq P_{0i}^{\pm} \leq P_{\max}$$

$$\sum_{i=1}^{m} wc_i^{\pm} - \sum_{i=1}^{m} Yc_i^{\pm} \leq qc^{\pm}$$

$$qc_i^{\pm} \le Q^{\pm}$$

$$wc_{i\ max} \ge wc_i^{\pm} \ge Yc_i^{\pm} \ge 0$$

where $F$ is the system benefits and $wc_i^{\pm}$ is the vector of first-stage decision variables. Since water demand (i.e., the first-stage decision variables) is often confirmed by decision makers according to previous empirical values, it would be altered by random events in actual water resources situations in planning periods, leading to first-stage benefits $\sum_{i=1}^{m} \left(G_i^{\pm} - P_{0i}^{\pm}\right) \times wc_i^{\pm}$. $Yc_i^{\pm}$ is the recourse at the second stage in response to events (e.g., present water shortages), which leads to the expected value of the second-stage penalties $\sum_{i=1}^{m} \left(G_i^{\pm} - P_{1i}^{\pm}\right) \times Yc_i^{\pm}$. $G_i^{\pm}$ is each water rights value; $P_{0i}^{\pm}$ is the expected price of each water right (which varies with the distributable water rights); $P_{1i}^{\pm}$ is the real price of each water right (current agricultural water price); $qc_i^{\pm}$ is the total water rights that agriculture can distribute; and $Q^{\pm}$ is the total water available for agriculture.

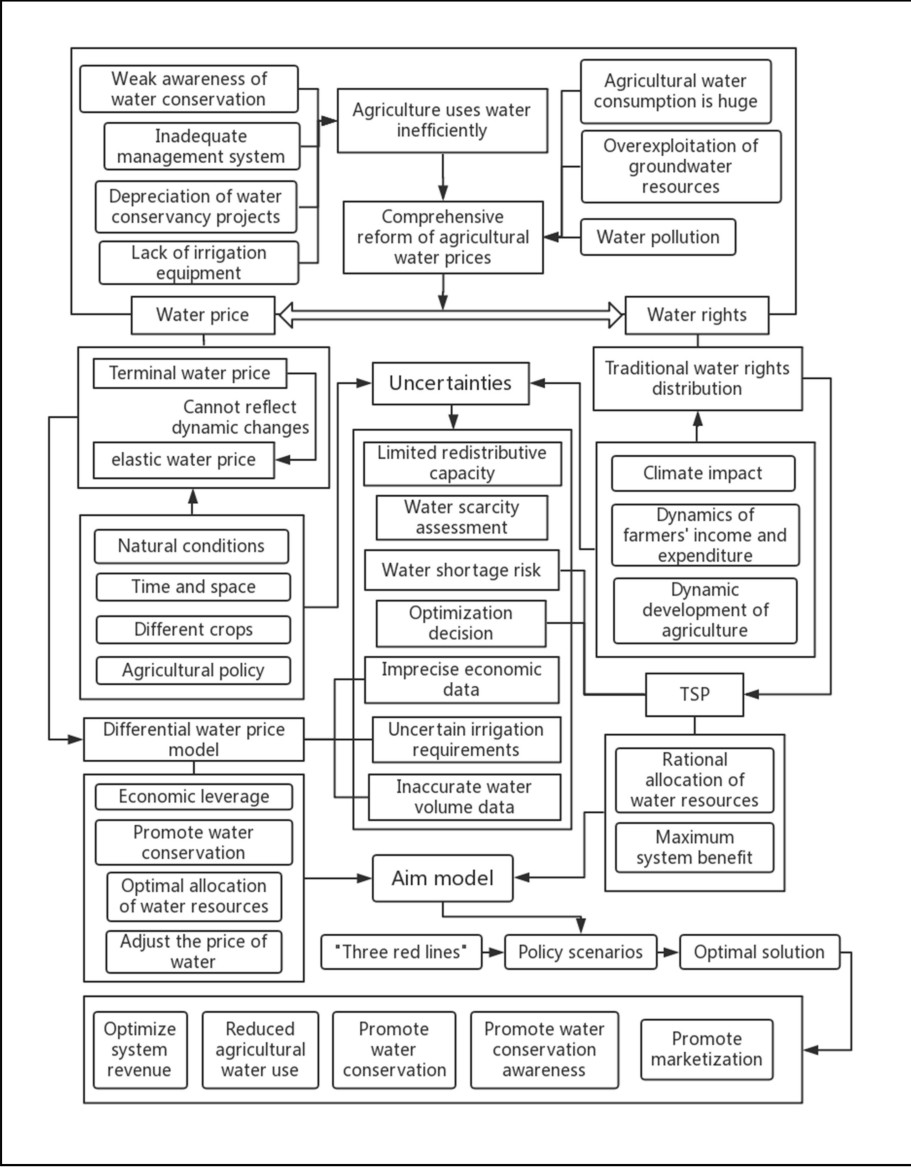

**Figure 1.** Framework of ITS-DWPM application in Hulin, China.

Water demand and the water price model were initialized by James and Lee:

$$Q = KP^{E_1}, \tag{2}$$

where $Q$ is the actual water consumption in irrigated areas; $K$ is constant; $P$ is the actual price of water; and $E_1$ is the coefficient of elasticity of water price. Mao (2005) took into account the influence of rainfall and evaporation in the rice growing season and built an agricultural water demand and water price model [12]:

$$qc = KP^{E_1}R^{E_2}Z^{E_3}, \tag{3}$$

where $P_{\max}$ is the maximum price that farmers can afford to pay for water; $R$ is the average annual evaporation during the growing season; $Z$ is the average annual rainfall during the growing season; $E_2$ is the coefficient of elasticity of evaporation; and $E_3$ is the coefficient of elasticity of rainfall. The theory is to add the regulation of water price into the initial allocation of water rights, so as to affect the distribution of water rights among various agricultural irrigation areas and at the same time ensure the maximization of system income, and finally to achieve a more optimal distribution of initial agricultural water rights [30–32]. Expected water price will change as the demand for water changes, so replace the expected water price with the changed water price.

$$\text{Max } f^{\pm} = \sum_{i=1}^{m}\left(G_i^{\pm} - P_{0i}^{\pm}\right) \times wc_i^{\pm} - \sum_{i=1}^{m}(G_i^{\pm} - P_{1i}^{\pm}) \times Yc_i^{\pm},$$

$$qc = KP_{0i}^{E_1}R^{E_2}Z^{E_3}$$

$$
\begin{aligned}
\Rightarrow \text{Max } f^{\pm} &= \sum_{i=1}^{m}\left(G_i^{\pm} - P_{0i}^{\pm}\right) \times wc_i^{\pm} - \sum_{i=1}^{m}(G_i^{\pm} - P_{1i}^{\pm}) \times Yc_i^{\pm} \\
&= \sum_{i=1}^{m}\left\{ G_i^{\pm} - (qc^{\pm}/KR^{E_2}Z^{E_3})^{\frac{1}{E_{1i}}} \right\} - \sum_{i=1}^{m}(G_i^{\pm} - P_{1i}^{\pm}) \times Yc_i^{\pm}
\end{aligned} \tag{4}
$$

subject to

$$P_{1i}^{\pm} \leq P_{0i}^{\pm} \leq P_{\max}$$

$$\sum_{i=1}^{m} wc_i^{\pm} - \sum_{i=1}^{m} Yc_i^{\pm} \leq qc^{\pm}$$

$$qc_i^{\pm} \leq Q^{\pm}$$

$$wc_{i\ \max} \geq wc_i^{\pm} \geq Yc_i^{\pm} \geq 0.$$

Then, an interval linear programming (ILP) solution is proposed for solving the interval two-stage programming (ITS) model. The objective function is not only a net profit, but also a 'punishment'. In the above planning, it is difficult to determine whether $wc_i^{+}$ should correspond to $f_{\text{opt}}^{+}$, and $wc_i^{-}$ should correspond to $f_{\text{opt}}^{-}$ (decision $wc_i^{+}$ will generate large economic benefits, but brings with it a lot of risk; $wc_i^{-}$ corresponds to a small economic benefit, but also small risk). Since each irrigation area has a fallow irrigation area every year, when decision $wc_i^{+}$ appears, excess water can irrigate these fields. Therefore, $wc_i^{+}$ corresponds to $f_{\text{opt}}^{+}$, and $wc_i^{-}$ should correspond to $f_{\text{opt}}^{-}$.

Solving steps:

$$(1)\ wc_i^{+} = wc_i^{-} + \Delta wc_i y_i,\ \Delta wc_i = wc_i^{+} - wc_i^{-};\ 0 \leq y_i \leq 1$$

$$\text{Max } f^{\pm} = \sum_{i=1}^{m}\left(G_i^{\pm} - P_{0i}^{\pm}\right) \times (wc_i^{-} + \Delta wc_i y_i) - \sum_{i=1}^{m}(G_i^{\pm} - P_{1i}^{\pm}) \times Yc_i^{\pm}$$

$$= \sum_{i=1}^{m}\left\{ G_i^{\pm} - (qc^{\pm}/KR^{E_2}Z^{E_3})^{\frac{1}{E_{1i}}} \right\} \times (wc_i^{-} + \Delta wc_i y_i) - \sum_{i=1}^{m}(G_i^{\pm} - P_{1i}^{\pm}) \times Yc_i^{\pm}$$

subject to

$$P_{1i}^{\pm} \leq P_{0i}^{\pm} \leq P_{\max}$$

$$Q^{\pm} \geq qc^{\pm} \geq \sum_{i=1}^{m} (wc_i^- + \Delta wc_i y_i - Yc_i^{\pm})$$

$$wc_{i\,max}^{\pm} \geq wc_i^- + \Delta wc_i y_i \geq Yc_i^{\pm} \geq 0, \forall i$$

$$0 \leq y_i \leq 1$$

$$\Rightarrow$$

(2)    This corresponds to $f_{opt}^+$:

$$\text{Max } f^+ = \sum_{i=1}^{m} \left(G_i^+ - P_{0i}^+\right) \times (wc_i^- + \Delta wc_i y_i) - \sum_{i=1}^{m} (G_i^- - P_{1i}^-) \times Yc_i^-$$
$$= \sum_{i=1}^{m} \left\{ G_i^+ - (qc^+ / KR^{E_2} Z^{E_3})^{\frac{1}{E_{1i}}} \right\} \times (wc_i^- + \Delta wc_i y_i) - \sum_{i=1}^{m} (G_i^- - P_{1i}^-) \times Yc_i^-$$

subject to

$$P_{1i}^- \leq P_{0i}^+ \leq P_{max}$$

$$qc^+ - \sum_{i=1}^{m} wc_i^- \geq \sum_{i=1}^{m} (\Delta wc_i y_i - Yc_i^{\pm}), \forall i$$

$$\Delta wc_i y_i \leq wc_{i\,max}^+ - wc_i^-, \forall i$$
$$Yc_i^- - \Delta wc_i y_i \leq wc_i^-, \forall i$$

$$Yc_i^- \geq 0, \forall i$$
$$0 \leq y_i \leq 1, \forall i$$

Solve for this linear program to find $f_{opt}^+$, $y_{iopt}$ and $Yc_{iopt}^-$. $y_{iopt}$ is given and $y_i$ is substituted.
(3) This corresponds to $f_{opt}^-$:

$$\text{Max } f^- = \sum_{i=1}^{m} \left(G_i^- - P_{0i}^-\right) \times (wc_i^- + \Delta wc_i y_i) - \sum_{i=1}^{m} (G_i^+ - P_{1i}^+) \times Yc_i^+$$
$$= \sum_{i=1}^{m} \left\{ G_i^- - (qc^- / KR^{E_2} Z^{E_3})^{\frac{1}{E_{1i}}} \right\} \times (wc_i^- + \Delta wc_i y_i) - \sum_{i=1}^{m} (G_i^+ - P_{1i}^+) \times Yc_i^+$$

subject to

$$P_{1i}^+ \leq P_{0i}^- \leq P_{max}$$

$$qc^- - \sum_{i=1}^{m} wc_i^- \geq \sum_{i=1}^{m} (\Delta wc_i y_i - Yc_i^{\pm}), \forall i$$

$$Yc_i^+ - \Delta wc_i y_i \leq wc_i^-, \forall i$$

$$Yc_i^- \geq Yc_{iopt}^-, \forall i$$

Solved for $f_{opt}^-$ and $Yc_{iopt}^+$.

Determine the optimal solution and the optimal value:

$$f_{opt}^{\pm} = \left[ f_{opt}^-, f_{opt}^+ \right]$$

$$Yc_{iopt}^{\pm} = \left[ Yc_{iopt}^-, Yc_{iopt}^+ \right], \forall i$$

For the water allocation problem corresponding to the initial model, the optimal allocation quantity can be determined:

$$A_{iopt}^{\pm} = wc_{iopt}^+ - Yc_{iopt}^+ = wc_i^- + \Delta wc_i y_{iopt} - Yc_{iopt}^{\pm}, \forall i$$

### 3. Case Study

Hulin City is located in the eastern part of Heilongjiang Province, at the southern foot of the Wanda Mountains and on the left bank of the Wusuli River. The geographical coordinates are 45°23′ to 46°36′ N and 132°11′ to 133°56′ E. The northwest of Hulin is bordered by Baolong County with Laolongbei and Jiangjun Ridge of the Wandashan tributary; the northeast is bordered by Qiliqin River and Raohe County; the west is bordered by Bailong Ridge and Mishan City, geographical location as shown in Figure 2.

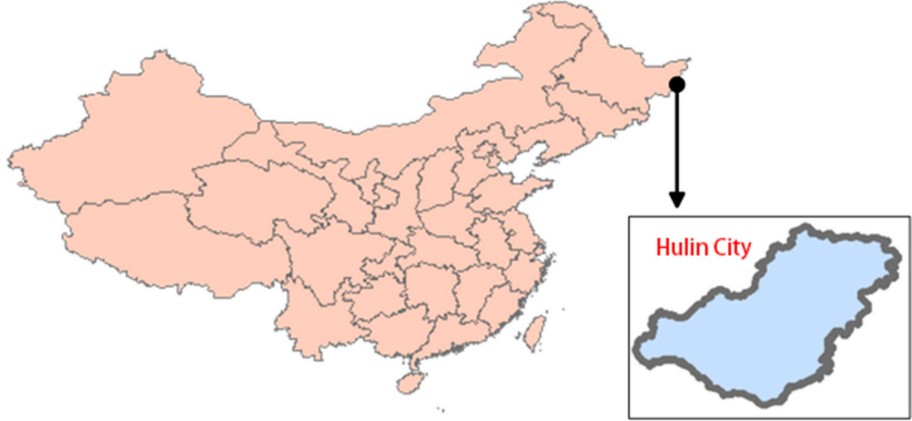

**Figure 2.** Study area.

For a long time, Heilongjiang Province has been a major agricultural province, but inadequate irrigation systems have led to farmers' use of irrigation water not being standardized. For example, local managers in some areas have not been able to collect water charges for a long time; farmers have directly pumped river water or pumped wells to extract groundwater without passing through management stations. Disordered irrigation often results in a lack of timely access to water in downstream irrigation areas. The exclusive crop in the study area was rice, and mostly surface water such as reservoirs and rivers was used for irrigation. In areas without surface runoff, groundwater is used for irrigation.

The price of the production of water in the Hulin area is 3.7 yuan/m$^3$: the price of domestic water is 2.6 yuan/m$^3$, while the price of agricultural water is only 0.062 yuan/m$^3$. The agricultural water price is too low, resulting in farmers' poor awareness of the necessity of saving water.

In order to establish a benign operating mechanism for agricultural water-saving and farmland water conservancy projects, in January 2016, the General Office of the State Council issued its "Opinions on Promoting Comprehensive Reforms of Agricultural Water Prices" to promote comprehensive reforms of agricultural water prices across the country.

The lack of water resources in Hulin City is an important factor restricting its social and economic development. The total amount of water available for agriculture in Hulin City is reduced, and the proportion of agricultural water consumption is high. Agricultural irrigation water is inexpensive and farmland water conservancy projects have seriously declined, causing this area to be a major source of agriculture water waste. Therefore, we should accelerate the comprehensive reform of agricultural water prices and improve the efficiency of agricultural water use. Raising the price of agricultural water will strengthen farmers' awareness of water conservation so that they can take measures to improve the efficiency of agricultural water use.

Lining main canals and branch canals is the most important measure to save water in agricultural irrigated areas. The irrigation water utilization coefficient has a logarithmic relationship with the lining rate of the canal. With the gradual increase in the lining rate, the irrigation water utilization coefficient increases less and less, so the result shows a trend of decreasing marginal efficiency. By studying the influence of channel lining on irrigation water utilization coefficient, the relationship

between channel lining rate and canal water utilization coefficient was developed [33]. The Hulin City field water utilization coefficient is 0.824, The lining rate of the canal is the independent variable, and the irrigation water utilization coefficient is the dependent variable. The quantitative relationship can be expressed as follows:

$$\eta = 0.366 + 0.68 \ln(\lambda - 2.748) \tag{5}$$

where $\eta$ is the utilization coefficient of irrigation water and $\lambda$ is the channel lining rate.

According to the engineering matching rate, controlled area, regional water resource conditions, and water-saving capacity, the net water quota of each irrigated area in the city can be controlled at 0.65 $m^3/m^2$. By 2020, the effective utilization coefficient of irrigation water will be raised to more than 0.55 (No. 1 document of the CPC Central Committee in 2011). By 2030, the effective utilization coefficient of irrigation water for farmland will be increased to more than 0.6, according to the National Comprehensive Plan for Water Resources.

The proposed ITS-DWPM method is considered to be applicable for tackling such a problem. Thus, we set up the following water-saving efficiency policy scenarios:

$$
\begin{aligned}
\text{Max } f^{\pm} &= \sum_{i=1}^{m} \left( G_i^{\pm} - P_{0i}^{\pm} \right) \times wc_i^{\pm} - \sum_{i=1}^{m} \left( G_i^{\pm} - P_{1i}^{\pm} \right) \times Yc_i^{\pm} + \sum_{i=1}^{m} T_i^{\pm} \times wo_i^{\pm} - \sum_{i=1}^{m} F_i^{\pm} \times Ye_i^{\pm} \\
&= \sum_{i=1}^{m} \left\{ G_i^{\pm} - (qc^{\pm}/KR^{E_2}Z^{E_3})^{\frac{1}{E_{1i}}} \right\} wc_i^{\pm} - \sum_{i=1}^{m} \left( G_i^{\pm} - P_{1i}^{\pm} \right) \times Yc_i^{\pm} + \\
&\quad \sum_{i=1}^{m} \left( P_{2i}^{\pm} + R_i^{\pm} \right) \times wo_i^{\pm} - \sum_{i=1}^{m} \left[ (\lambda_{0i}^{\pm} - \lambda_{1i}^{\pm}) \times P_{1i}^{\pm} \right] \times Ye_i^{\pm},
\end{aligned} \tag{6}
$$

subject to

$$P_{1i}^{\pm} \leq P_{0i}^{\pm} \leq P_{\max}$$
$$\lambda_{0i}^{\pm} \geq \lambda_{1i}^{\pm}$$

$$\sum_{i=1}^{m} wc_i^{\pm} - \sum_{i=1}^{m} Yc_i^{\pm} \leq qc^{\pm}$$

$$qc_i^{\pm} \leq Q^{\pm}$$

$$wc_{i \, \max} \geq wc_i^{\pm} \geq Yc_i^{\pm} \geq 0$$

$$P_{1i}^{\pm} \leq P_{0i}^{\pm} \leq P_{\max}$$
$$\lambda_{0i}^{\pm} \geq \lambda_{1i}^{\pm}$$

where $T_i^{\pm}$ is the benefit of each water conservation right; $wo_i^{\pm}$ is the anticipated water conservation rights.; $F_i^{\pm}$ is the cost savings for each water right; $Ye_i^{\pm}$ is the number of water rights after optimized allocation in the second stage; $P_{2i}^{\pm}$ is the expected water price; $R_i^{\pm}$ is the water-saving reward; $\lambda_{0i}^{\pm}$ is the expected rate of lining of agricultural irrigation channels; and $\lambda_{1i}^{\pm}$ is the real rate of lining of agricultural irrigation channels.

(1)    Agricultural water distribution

According to the above analysis of the overall ideas and basic conditions for agricultural water use rights, it is clear that agricultural water distribution and paddy field irrigation area are two key factors. The area of irrigation of the paddy field determines the proportion of water allocated to each irrigation district. Taking the total water control of "three red lines" in Hulin City as the target, after deducting the domestic water, ecological water, nonagricultural production water, and reserved water, the remaining water will be used for agricultural water distribution in Hulin City. It is calculated as follows:

$$W_a = W_r - W_l - W_e - W_i - W_p, \tag{7}$$

where $W_a$ is the total agricultural water distribution; $W_r$ is Hulin City's 2017 "Three Red Lines" total water control indicators; $W_l$ is the life water distribution; $W_e$ is the ecological water distribution; $W_i$ is the industry water distribution; and $W_p$ is the reserved water distribution.

The total water control indicators of the "three red lines" in Hulin City are $397 \times 10^6$ m$^3$ of surface water and $365 \times 10^6$ m$^3$ of groundwater, totaling $762 \times 10^6$ m$^3$. According to the accounting of agricultural water distribution in Hulin City, the amount of water that can be distributed in agriculture is $713 \times 10^6$ m$^3$, of which the surface water is $377 \times 10^6$ m$^3$, and the groundwater is $336 \times 10^6$ m$^3$.

(2)  Agricultural water rights allocation

According to the Heilongjiang Provincial Farmland Water Conservancy Management Station's 2015 Heilongjiang Paddy Field Development Report, compiled by the Heilongjiang Provincial Farmland Water Management Center, we adjusted the effective area of large-scale irrigation according to the data provided by the Hulin City Irrigation District Management Station. The amount of water that can be allocated for agriculture is calculated based on the accounting results, as shown in Table 1 for the expected water rights threshold.

**Table 1.** Hulin water calculation [34].

| Water Category | Surface Water ($10^6$ m$^3$) | Groundwater ($10^6$ m$^3$) | Total ($10^6$ m$^3$) |
|---|---|---|---|
| "Three Red Lines" | 39,700 | 36,500 | 76,200 |
| Life | | 613.71 | 613.71 |
| Ecology | 13.87 | 44.92 | 58.79 |
| Industry | | 415.81 | 415.81 |
| Reserve | 1985 | 1825 | 3810 |
| Agriculture | 37,701.13 | 33,600.56 | 71,301.69 |

From Table 2, the distribution of agricultural water use in Hulin City and the actual verification, the amount of water allocated to large-scale irrigation districts in Hulin City is $171.85 \times 10^6$ m$^3$. According to the data provided by the Hulin City Irrigation District Management Station, there are five large-scale irrigation districts in the county, and the current water price of agricultural products in Hulin City is 0.062 yuan/m$^3$.

**Table 2.** Expected water demand target ($10^6$ m$^3$) [34].

| User | Total Control Policy Scenario | | | |
|---|---|---|---|---|
| | 3% | 5% | 10% | 15% |
| Hulin | (54, 55) | (58, 60) | (40, 45) | (48, 50) |
| Hutou | (13, 16.5) | (11, 15.75) | (12, 16) | (8, 10) |
| Shitouhe | (12.5, 16) | (11, 14.94) | (12, 15.5) | (8, 10) |
| Daxinancha | (30, 34) | (30, 35) | (29, 33.5) | (23.5, 26) |
| Abei | (30, 33) | (28, 30) | (31.45, 36.65) | (23, 25) |
| Other | (477, 480) | (450, 455) | (477, 482) | (377, 380) |

(3)  Economic data

In 2016, the total GDP of Hulin City was 13,186.78 $\times$ 10$^6$ yuan, of which the primary, secondary, and tertiary industries' added value were 7983.92 $\times$ 10$^6$ yuan, 1709.73 $\times$ 10$^6$ yuan, and 3493.13 $\times$ 10$^6$ yuan, respectively. In 2016, the total population of Hulin City reached 281,114, including 195,342 urban residents and 85,772 rural residents. In 2016, the per capita disposable income of urban residents in Hulin reached 23,067 yuan, and the per capita disposable income of rural residents reached 16,462 yuan [35,36]. Table 3 shows the unit water rights gains and losses under different policy scenarios.

**Table 3.** Economic data [36,37].

| Total Control Policy Scenario | User | | | | | |
|---|---|---|---|---|---|---|
| | Hulin | Hutou | Shitouhe | Daxinancha | Abei | Other |
| Unit water rights benefit(yuan/m$^3$) | | | | | | |
| 3% | (14.68, 15.30) | (6.09, 6.35) | (6.13, 6.39) | (6.09, 6.35) | (6.97, 7.26) | (10.21, 10.64) |
| 5% | (14.68, 15.30) | (6.09, 6.35) | (6.13, 6.39) | (6.09, 6.35) | (6.97, 7.26) | (10.21, 10.64) |
| 10% | (14.68, 15.30) | (6.09, 6.35) | (6.13, 6.39) | (6.09, 6.35) | (6.97, 7.26) | (10.21, 10.64) |
| 15% | (14.68, 15.30) | (6.09, 6.35) | (6.13, 6.39) | (6.09, 6.35) | (6.97, 7.26) | (10.21, 10.64) |
| Unit water rights shortage(yuan/m$^3$) | | | | | | |
| 3% | (15.61, 16.23) | (6.48, 6.73) | (6.52, 6.78) | (6.48, 6.73) | (7.41, 7.70) | (10.86, 11.29) |
| 5% | (15.61, 16.23) | (6.48, 6.73) | (6.52, 6.78) | (6.48, 6.73) | (7.41, 7.70) | (10.86, 11.29) |
| 10% | (15.61, 16.23) | (6.48, 6.73) | (6.52, 6.78) | (6.48, 6.73) | (7.41, 7.70) | (10.86, 11.29) |
| 15% | (15.61, 16.23) | (6.48, 6.73) | (6.52, 6.78) | (6.48, 6.73) | (7.41, 7.70) | (10.86, 11.29) |

Hulin has a temperate continental monsoon climate and is in a mild and humid climate zone in the Sanjiang Plain. Cold and snowy in winter; summers are short, warm, and rainy; windy in spring; autumns are rainy and cool. The annual average temperature is 3.5 °C, with the coldest temperatures in January. The monthly average temperature in January is −18.3 °C, while the extreme minimum temperature recorded was −36.1 °C. The weather is hottest in July, with a monthly average temperature of 21.6 °C and an extreme maximum temperature of 35.2 °C. The annual average rainfall amount is 566 mm, the precipitation in the growing season (May to September) can account for 80–90% of the total amount in a year, and the regional humidification coefficient is 0.7. Hulin City is located in Heilongjiang Province, and the rainfall is mostly in the growing season, which is 85% of the year. The annual evaporation is basically only in the growing season.

## 4. Results and Discussion

### 4.1. Results Analysis

In 2012, the State Council issued its "Opinions on Implementing the Strict Water Resources Management System," clearly proposing the main objectives of water resources development and utilization control, water efficiency control, and water function zones. Since the implementation of the most stringent water system in Heilongjiang Province, the total amount of water used has decreased by 2.7% from 2013 to 2016 [36]. Therefore, in response to the plan for total agricultural water use control in Hulin City by 2020, the total agricultural water consumption in Hulin City will be reduced by 3%. In 2022, the total agricultural water consumption in Hulin City will be reduced by 5%. In 2025, the total water consumption will be reduced by 10%, and by 2030, the total water consumption of Hulin City will be reduced by 15%. According to the policy of setting water resources control for water use efficiency in Hulin City, by 2020 the utilization coefficient of agricultural irrigation water in Hulin City will reach 0.55 or above (2011 No. 1 Document). In 2030, the utilization coefficient of agricultural irrigation water in Hulin City will reach 0.60 or above, according to the National Comprehensive Plan for Water Resources.

The reduction of agricultural irrigation water in each irrigation area is shown in Table 4. The reduction in the Other irrigation area is the largest, while the reduction of agricultural irrigation water for Hutou and Shitouhe irrigation areas is the lowest. The Other irrigation area is the largest area, so the water consumption and corresponding irrigation water consumption are also the largest. By contrast, the Hutou and Shitouhe irrigation areas are small, so the total water consumption is the lowest and the corresponding reduction is the least.

**Table 4.** Optimized water targets under scenario ($10^6$ m$^3$).

| Total Control Policy Scenario | User | | | | | |
|---|---|---|---|---|---|---|
| | Hulin | Hutou | Shitouhe | Daxinancha | Abei | Other |
| | $y_i$ | | | | | |
| 3% | 1 | 0.13 | 0.12 | 0.45 | 1 | 1 |
| 5% | 1 | 0.47 | 0.43 | 0.23 | 1 | 1 |
| 10% | 1 | 0.13 | 0.01 | 0.11 | 0.17 | 1 |
| 15% | 1 | 1 | 1 | 1 | 1 | 1 |
| | Water consumption reduction in each irrigation area ($10^6$ m$^3$) | | | | | |
| 3% | (1.65, 2.35) | (0.64, 0.84) | (0.57, 0.78) | (1.50, 2.55) | (0.99, 3.75) | (14.40, 14.93) |
| 5% | (3.00, 4.32) | (0.66, 2.67) | (0.64, 2.14) | (1.56, 2.34) | (1.50, 3.12) | (22.75, 23.00) |
| 10% | (5.40, 9.80) | (1.50, 1.98) | (1.44, 1.47) | (3.54, 3.98) | (3.88, 4.67) | (57.84, 62.24) |
| 15% | (11.50, 13.52) | (2.30, 3.92) | (2.30, 3.92) | (5.98, 8.14) | (5.75, 7.52) | (87.4, 93.48) |
| | Optimal target under the scenario ($10^6$ m$^3$) | | | | | |
| 3% | 71.55 | 12.74 | 12.32 | 29.89 | 32.02 | 506.68 |
| 5% | 68.42 | 12.02 | 11.75 | 29.14 | 31.45 | 487.62 |
| 10% | 62.70 | 11.03 | 10.59 | 25.96 | 28.40 | 422.60 |
| 15% | 52.90 | 9.54 | 9.06 | 21.88 | 24.78 | 366.79 |

It can be seen from Figure 3 that the water consumption of Hutou, Shitouhe, Daxinancha, and Abei is small, but the corresponding proportion is relatively high. From the results, the reduction ratio is inversely proportional to the unit water rights income, while the reduction ratio is directly proportional to the unit water rights loss, which is in line with actual expectations. The increase of the Hutou, Shitouhe, and Daxinancha ratios indicates that the water management of irrigation districts and farmland water conservancy projects still needs to be improved.

The water price of agricultural irrigation districts in Hulin City adopts the unified pricing method. The water price of agricultural water in irrigation districts under different policy scenarios is shown in Figure 3.

From Figure 4, we can see the effect of economic leverage on the price of agricultural irrigation water. Under different policy scenarios, the change in agricultural water consumption is also in line with objective economic laws. A reasonable rise in the water price can effectively promote water conservation in agriculture. The initial water rights model of differential water prices can fully consider the economic impact of water price changes in the water rights optimization process. Similarly, considering that the irrigation water utilization coefficient of each agricultural irrigation area should reach 0.55 in 2020, a reasonable agricultural water price should be formulated that fully considers the actual water use situation and the improvements that can be achieved by engineering measures in Hulin City, to accordingly formulate reasonable agricultural water-saving policies. The current low price of agricultural water has led to a lack of water-saving awareness among water users in irrigation districts, which ultimately leads to the waste of agricultural irrigation water and the elevation of water prices. It is also an important part of the comprehensive reform of agricultural water prices.

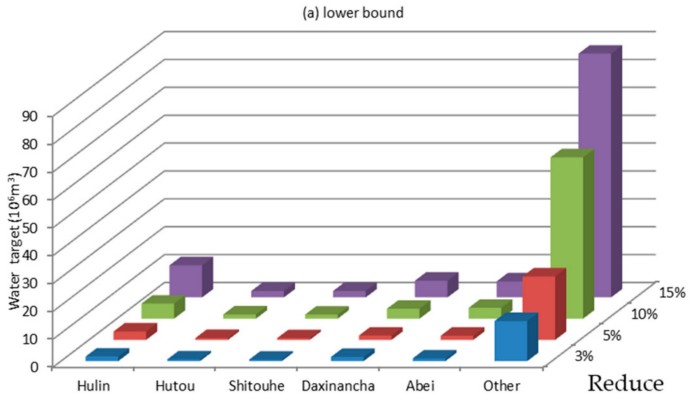

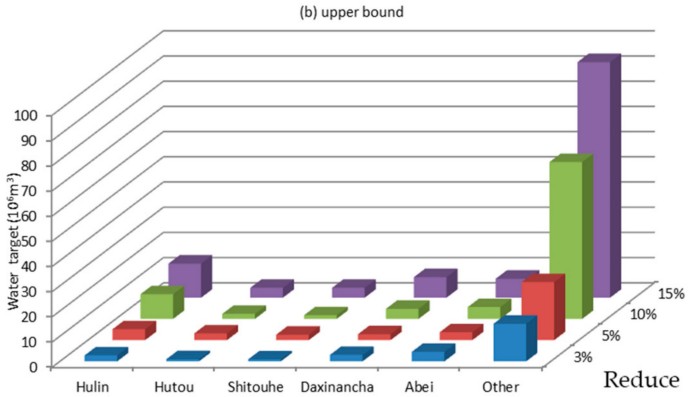

**Figure 3.** Optimized water targets under scenario. The data come from the model generation results (Table 4).

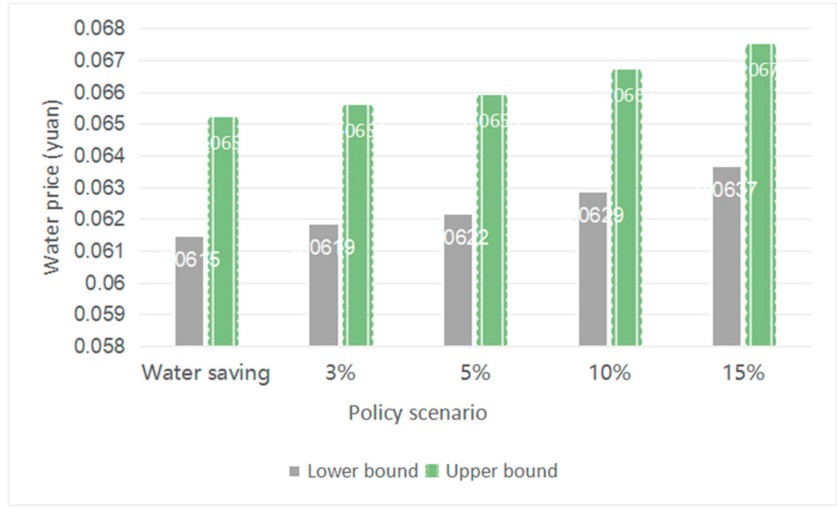

**Figure 4.** Hulin City agricultural differential water price. The data come from the model generation results (Table 4).

Figure 5 shows the differences in water use efficiency between irrigation districts. Shitouhe has the lowest water efficiency and Hulin has the highest water efficiency. Water consumption reduction in the irrigation area is not a one-off. When the water consumption is reduced to a certain extreme value, it will generate additional and greater losses. Therefore, under the 15% policy scenario, the Hutou,

Shitouhe, Daxinancha, and Abei water reductions are only based on their lower bounds, and the remaining reductions are allocated to Hulin and Other.

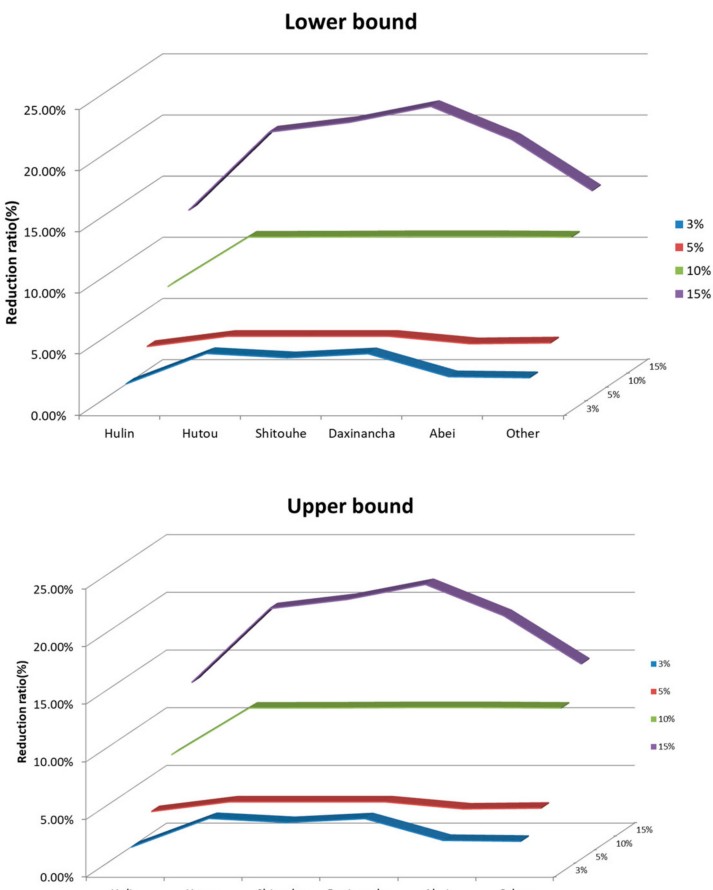

**Figure 5.** Agricultural irrigation water rights reduction ratio. The data come from the model generation results (Table 4).

Due to the randomness of the results of various accidental factors during irrigation, the irrigation water consumption in the irrigation district is standardized by the upper and lower boundaries of the water rights redistribution in the irrigation district. The optimal allocation interval of agricultural water use in each irrigation district is shown in Figure 6.

Figure 7 shows that only a small percentage of users are water inefficient. As the proportion of agricultural water is reduced, users with low water use efficiency will cause huge losses to users when the proportion of water rights is reduced to a certain extent. At this point, the amount of the reduction is allocated to the higher-yield users.

According to Table 4, in 2020, the utilization coefficient of agricultural irrigation water in Hulin City increased to 0.55. Hutou needs to increase agricultural irrigation channels by at least 4.49%; after considering the cost of water saving and water-saving income, the total system revenue will be reduced by $0.9 \times 10^3$ yuan. Shitouhe needs to increase the agricultural irrigation channel by at least 4.03%, while the total system revenue is reduced by $0.5 \times 10^3$ yuan. Daxinancha needs to increase the agricultural irrigation channel by at least 4.49%, while the total system revenue is reduced by $2.5 \times 10^3$ yuan. For irrigation districts, because the water-saving cost is relatively higher than the water-saving income, the water-saving enthusiasm of the corresponding irrigation districts will be greatly reduced, which requires the government to develop effective and reasonable water-saving reward and compensation mechanisms.

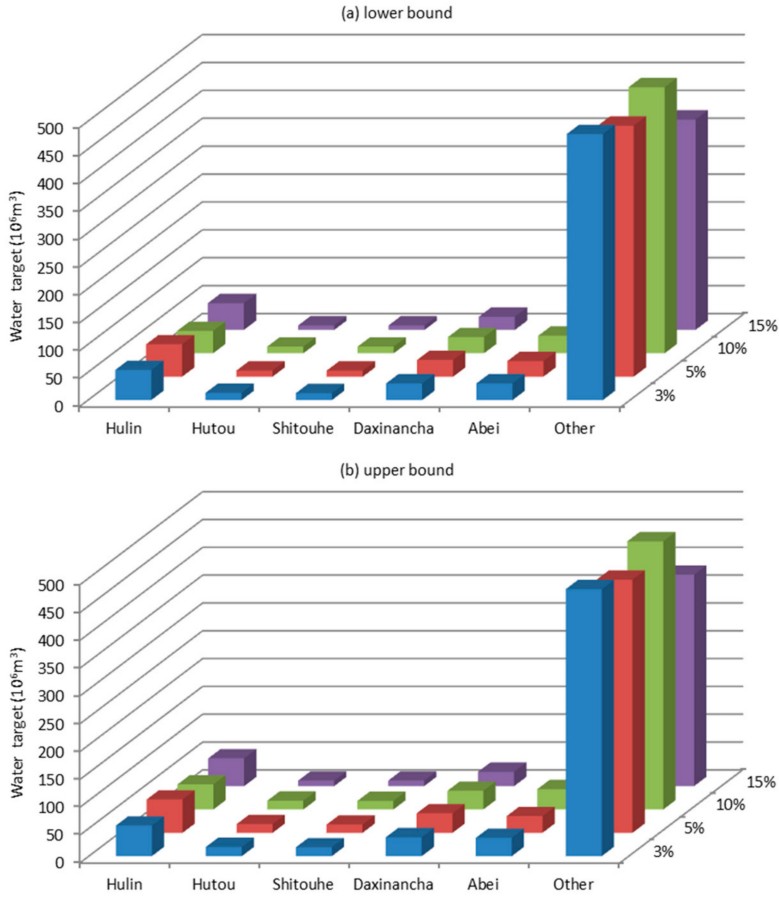

**Figure 6.** Water rights reduction under scenario. The data come from the model generation results (Table 4).

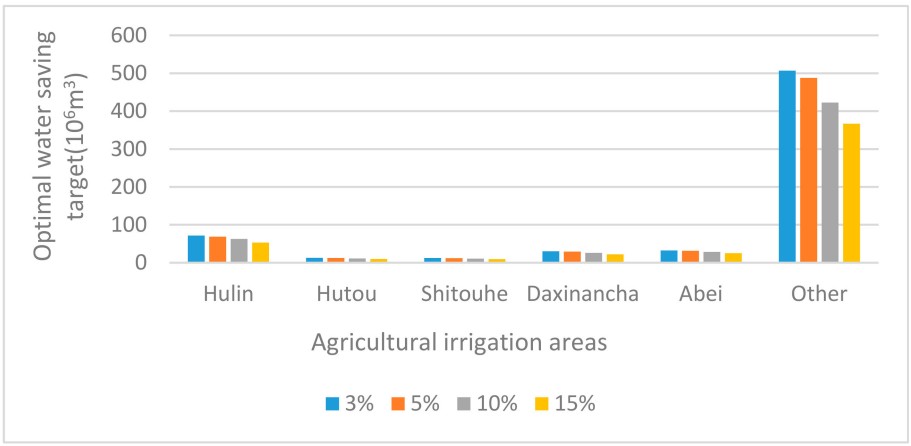

**Figure 7.** Optimal target of water rights allocation.

From Table 5, it is recommended to develop water-saving incentives as follows: The water-saving reward fund is directly awarded to the water user by the irrigation district management unit. The water-saving capacity is between 20% and 50%. According to the difference between the total water consumption index and the actual water consumption, the reward is 0.03 yuan/m$^3$. The water-saving amount exceeds 50%, and the difference between the total water consumption index and the actual water consumption is 0.045 yuan/m$^3$.

**Table 5.** Water saving targets.

| Zone | User | | | | | |
|------|------|------|------|------|------|------|
| | **Hulin** | **Hutou** | **Shitouhe** | **Daxinancha** | **Abei** | **Other** |
| Lower bound | 0 | 0.58 | 0.50 | 1.36 | 0 | 0 |
| Upper bound | 0 | 0.64 | 0.57 | 1.50 | 0 | 0 |

According to the results of this study for Hutou, Shitouhe, and Daxinancha, the government should formulate corresponding compensation measures based on the water-saving benefits of each irrigation district, and the threshold for compensation should not be too high. The model also provides relevant scientific guidance for the government to formulate more detailed reward measures. The government will formulate reasonable reward and subsidy measures for water saving in irrigation districts, which can effectively encourage irrigation districts to manage irrigation water more carefully and increase the degree of investment. Water users can have greater water awareness while ensuring maximum water use revenue in irrigation districts.

*4.2. Discussion*

In this study, an ITS-DWPM has been developed for initial water rights allocation in agricultural water rights work. It has the following benefits: (i) It can reflect the tradeoffs between the predefined economic targets and the associated water shortage penalties/satisfied water gain, as well as the fuzziness of the water availability (i.e., fuzzy manner); (ii) it can handle the uncertainty of data sources, as well as the fuzziness of the water availability; (iii) limited economic data can be expressed as interval numbers, which would be acceptable as its uncertain inputs; (iv) it can handle nonlinearity in cost/benefit objectives, and has a global optimum under a number of system conditions.

The application of ITS-DWPM in the initial water rights allocation of agricultural irrigation districts in Hulin City can effectively accelerate the process of agricultural water rights work. It is worth mentioning that: (a) the water use efficiency in the Hutou, Shitouhe, and Daxinancha irrigation areas is low, and the corresponding initial water rights allocation adjustment will strengthen water users' awareness of water conservation and improve their water use efficiency. Therefore, the adjustment of the initial water rights allocation is conducive to maximizing the system revenue. (b) A more complete irrigation system can effectively promote agricultural water conservation, and refined management of irrigation districts will be a necessary condition for future irrigation districts. Therefore, the rational formulation of irrigation systems also relies on multi-stage scientific water resources planning. (c) Uncertainty in the data will adversely affect the results, while improvements in irrigation infrastructure will improve the conditions, which would make water allocation easier. The ITS-DWPM can simultaneously deal with nonlinear and interval features in the initial water rights allocation. The optimal allocation of initial water rights can greatly reduce the waste of water resources, particularly in developing regions.

## 5. Conclusions

In this study, an inexact two-stage initial water rights allocation (ITS) model has been developed that combines the approaches of two-stage stochastic programming (TSP) and interval-parameter programming (IPP). Meanwhile, it can resolve imprecise economic data and non-linear in DWPM. The main research contents and policy recommendations are as follows:

(1)    Main research contents

This study considers the uncertainty and complexity in the water resources planning system to ensure the maximum return of the system while considering the dynamic changes of various factors. Taking the market influence and economic leverage adjustment as the preconditions, with total agricultural water control and efficiency as the goals, we constructed an initial water rights

optimization allocation model that incorporated the changes in differentiated water prices. Continuous optimization of empirical decision-making by a two-stage planning method, based on the actual water-saving target of the agricultural irrigation district in Hulin City, led to an initial water rights allocation model (ITS-WRIDM) based on differential water prices for water-saving adjustment in various agricultural irrigation districts in Hulin City. The developed model reflects dynamic water price changes and real-time changes in agricultural policies, as well as the multiple uncertainties in the process of establishing and solving the optimization model, which makes the initial water rights allocation in the agricultural irrigation district more reasonable.

(2)   Policy recommendations

(a) When the total amount of agricultural water is carefully controlled and users' water-saving awareness is encouraged, the water rights reduction of each irrigation area can reach 20%, which is an extreme value; continued reduction will have a great impact on agricultural production and the social economy. (b) When agricultural water efficiency is controlled, the agricultural irrigation water utilization coefficient will eventually reach a certain extreme value. At this time, increasing investment will cause economic waste. In 2020, the irrigation water utilization coefficient of each agricultural irrigation area in Hulin City should reach 0.55. Hutou needs to increase its agricultural irrigation channels by at least 4.49%, Shitouhe by at least 4.03%, and Daxinancha by at least 4.49%. (c) The imperfection of the current agricultural water-saving reward mechanism will lead to a significant reduction in water-saving enthusiasm in each irrigation district. Water-saving rewards can be set up according to the amount of water saving. The higher the water-saving amount, the higher the unit's water-saving reward; however, the reward threshold should not be set too high if it is to be an efficient water-saving and compensation mechanism. The government can formulate the corresponding compensation measures according to the actual water-saving benefit of the irrigation district and the actual local economic level. (d) While the government strengthens macro-control, it should also consider the laws of the market economy, promotes the optimal allocation of water resources, and improves the efficiency of water resources utilization. The implementation of the agricultural water rights confirmation system must not only follow a reasonable initial water rights optimization allocation system, but also a reasonable water intake system and irrigation system.

**Author Contributions:** The main text was written by S.Y. L.W. was in charge of polishing the English. T.L. was responsible for preparing the basic data. Conceptualization, S.Y.; Data curation, S.Y. Formal analysis, S.Y. Funding acquisition, L.W.; Investigation, S.Y.; Methodology, S.Y.; Project administration, T.L.; Software, S.Y.; Validation, T.L.; Writing—original draft, S.Y.; Writing—review and editing, L.W. All authors have read and agreed to the published version of the manuscript.

**Funding:** This research was funded by the Self-designed Project of Heilongjiang Instituted of Water Conservancy Science, grant number (ZN) 201806. Heilongjiang Province Applied Technology Research and Development Guide Project (GZ16B013).

**Acknowledgments:** This research was supported by the Self-designed Project of Heilongjiang Instituted of Water Conservancy Science (ZN), 201806. The authors are grateful to the editors and the anonymous reviewers for their insightful comments and suggestions.

**Conflicts of Interest:** We declare that we do not have any commercial or associative interest that represents a conflict of interest in connection with the work submitted.

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
