# Peer review of "An Agricultural Interval Two-Stage Fuzzy Differential Water Price Model (ITS-DWPM) for Initial Water Rights Allocation in Hulin, China"

_water, doi:10.3390/w12010221_

Round 1

Reviewer 1 Report

The paper deals with a very relevant problem of water management, in the Heilongjiang province, China.
Authors claim that current water rights are not allocated efficiently, therefore causing overexploitation and waste of the resource.
The paper seems very promising, but there are some relevant flaws, requiring efforts for improvement.
As follows, I will list some of the improvement which are needed:
1. the paper need a proofreading by an English native speaker, who must also be aknowledged in Economics and/or resource management. In present form,
although it is generally understandable, the paper is not clear from the scientific point of view.
2. after the Introduction, the paper continues with the methodology, full of formulae, which are not connected with some basic economic concepts,
which are described in the following section. I strongly recommend a significant revision of the structure of the paper. After the Introduction,
the basic assumptions and hypotheses must be explained; I guess they are already in the paper, but in the wrong place (e.g. the current 3.1).
The mathematical formulae of section 2 must be combined with those reported on page 9.
3. Authors must clarify why at present, water use rights are not efficient. I do not believe that by simply charghing a higher price to poor farmers will
be the solution of all problems. In addition, as you mentioned in the paper, how is it possible to find an efficient solution if you do not have
reliable data about water supply and water demand?
4. Depending on crops, water consumption may be very different. For instance, in case of paddy fields, the amount of water consumption is very high.
However, in some irrigation systems (e.g. terrace paddy fields) water flows downstream, implying different re-use, for different farmers. On the
contrary, in more arid environment, water volumes are available to only one crop. Please, add the description of cropping patterns and some basic
information of irrigation systems (e.g. groundwater, streams, rivers, lakes, aqueducts, etc.). A thematic map will be helpful to explain
the current situation to foreign readers.
5. In Results, I have not seen the impacts of water reduction on farmers' income. As you mentioned in the paper, farmers are very poor.
Are you sure that charging a higher water price is the solution to save water? What are the assumptions and hypotheses behind this research?
Please, provide a clear explanation about all possible causes of inefficiencies which may be caused by a low water pricing, and explain why
your model may provide a solution, without robust data on water availability and water consumption.

Author Response

Point 1: the paper need a proofreading by an English native speaker, who must also be aknowledged in Economics and/or resource management. In present form, although it is generally understandable, the paper is not clear from the scientific point of view.

Response 1: I used MDPI-recommended editors to help me check the grammar and proofread the article in my native language.

Point 2: after the Introduction, the paper continues with the methodology, full of formulae, which are not connected with some basic economic concepts, which are described in the following section. I strongly recommend a significant revision of the structure of the paper. After the Introduction, the basic assumptions and hypotheses must be explained; I guess they are already in the paper, but in the wrong place (e.g. the current 3.1). The mathematical formulae of section 2 must be combined with those reported on page 9.

Response 2: I took your suggestion and adjusted the structure of the article.

Point 3: Authors must clarify why at present, water use rights are not efficient. I do not believe that by simply charghing a higher price to poor farmers will be the solution of all problems. In addition, as you mentioned in the paper, how is it possible to find an efficient solution if you do not have reliable data about water supply and water demand?

Response 3: In the case study, I added a table of water resources that can be allocated to agriculture in Hulin City, starting from the "three red lines" total water consumption control target in Hulin City.

Point 4: Depending on crops, water consumption may be very different. For instance, in case of paddy fields, the amount of water consumption is very high. However, in some irrigation systems (e.g. terrace paddy fields) water flows downstream, implying different re-use, for different farmers. On the contrary, in more arid environment, water volumes are available to only one crop. Please, add the description of cropping patterns and some basic information of irrigation systems (e.g. groundwater, streams, rivers, lakes, aqueducts, etc.). A thematic map will be helpful to explain the current situation to foreign readers.

Response 4: I added thematic maps in the case study, the agricultural irrigation situation in Hulin City.

Point 5:  In Results, I have not seen the impacts of water reduction on farmers' income. As you mentioned in the paper, farmers are very poor. Are you sure that charging a higher water price is the solution to save water? What are the assumptions and hypotheses behind this research? Please, provide a clear explanation about all possible causes of inefficiencies which may be caused by a low water pricing, and explain why your model may provide a solution, without robust data on water availability and water consumption.

Response 5: It is precisely because farmers' income is not high, so when raising the water price, they must also formulate reasonable incentive measures. I also gave reasonable incentive measures in the results analysis.

The comprehensive reform of agricultural water price is a systematic project, and its important feature is to co-ordinate the establishment of mechanisms for water management, engineering construction and management, precise subsidies for agricultural water prices, and incentives for saving water. Through the mechanism, it can realize the role of promoting water saving in agriculture and guarantee the construction and maintenance of the project.

China's water resources are very scarce and agricultural water accounts for a high proportion. However, agricultural water resources use efficiency is not high. Extensive and wasteful water problems are common, and flood irrigation methods are still common. The comprehensive reform of agricultural water prices has adopted facilities to save water, agronomic water, and management to save water. Under the conditions of scarce water resources, it is conducive to using less water resources to produce more and better food and improve food production security capabilities. The effectiveness in this regard has been proven during the implementation of water-saving and food-increasing in Northeast China.

One of the goals of the comprehensive reform of agricultural water prices is to give full play to the leverage of water prices, guide farmers to save water, promote the improvement of engineering facilities, and optimize and adjust the agricultural planting structure. Especially in terms of structural adjustment, it can promote moderate reduction of high water-consuming crop areas such as over-utilization of surface water and severe over-exploitation of groundwater, breed and promote drought-tolerant and water-saving crops that require less water, establish crop growth stages and natural Precipitation matches the agricultural planting structure and planting system.

As a result of my water-saving model, the policy recommendations are given in the conclusion part of the article. The best water-saving method for agricultural irrigation is to increase the canal lining. I give specific suggestions and the benefits brought by the conclusion part.

Reviewer 2 Report

      It would be desirable to include a specific heading related to the use of stochastic models in decision making for price management and allocation of quotas for agriculture, noting empirical advances in different territorial contexts, their advantages and the main obstacles, resistances and disadvantages that they represent your application.   As well as some references of international literature that enrich and complete the state of the art, defining more widely the theoretical framework that serves as a reference to the study.
  It would be desirable to highlight some limitations of the study, as well as some feasibility analysis that includes aspects related to cultural and contextual resistances for its implementation.

Author Response

Response: The background of the article is the policy of comprehensive reform of agricultural water prices. The source of the policy has been mentioned and increased.

Reviewer 3 Report

Indicate in the abstract the results of the case study.

It would be interesting to include in the discussion a comparison with the results of other authors.

Author Response

I added case results to the abstract.

comparison of results with other authors in discussions.

This manuscript is a resubmission of an earlier submission. The following is a list of the peer review reports and author responses from that submission.